# Mathematical Modelling Based on In Vivo Imaging Suggests CD137-Stimulated Cytotoxic T Lymphocytes Exert Superior Tumour Control Due to an Enhanced Antimitotic Effect on Tumour Cells

**DOI:** 10.3390/cancers13112567

**Published:** 2021-05-24

**Authors:** Richard J. Beck, Bettina Weigelin, Joost B. Beltman

**Affiliations:** 1Leiden Academic Centre for Drug Research, Division of Drug Discovery and Safety, Leiden University, 2333 CC Leiden, The Netherlands; r.j.beck@lacdr.leidenuniv.nl; 2Department of Preclinical Imaging and Radiopharmacy, Eberhard Karls University of Tübingen, 72076 Tübingen, Germany; Bettina.Weigelin@med.uni-tuebingen.de; 3Cluster of Excellence iFIT (EXC 2180) “Image-Guided and Functionally Instructed Tumor Therapies”, University of Tübingen, 72076 Tübingen, Germany

**Keywords:** mathematical modelling, ordinary differential equations, cytotoxic T lymphocytes, T cells, tumour, immunotherapy, CD137

## Abstract

**Simple Summary:**

Cytotoxic T lymphocytes (CTLs) play an important role in controlling tumours, and an improved understanding of how they accomplish this will benefit immunotherapeutic cancer treatment strategies. Stimulation of CTLs by targeting their CD137 receptor is a strategy currently under investigation for enhancing responses against tumours, yet so far only limited quantitative knowledge regarding the effects of such stimulation upon CTLs has been obtained. Here, we develop mathematical models to describe dynamic in vivo two-photon imaging of tumour infiltrating CTLs, to characterise differences in their function either in the presence or absence of a CD137 agonist antibody. We showed that an increased antiproliferative effect and a more sustained presence of CTLs within the tumour were the most important effects associated with anti-CD137 treatment.

**Abstract:**

Several immunotherapeutic strategies for the treatment of cancer are under development. Two prominent strategies are adoptive cell transfer (ACT) of CTLs and modulation of CTL function with immune checkpoint inhibitors or with costimulatory antibodies. Despite some success with these approaches, there remains a lack of detailed and quantitative descriptions of the events following CTL transfer and the impact of immunomodulation. Here, we have applied ordinary differential equation models to two photon imaging data derived from a B16F10 murine melanoma. Models were parameterised with data from two different treatment conditions: either ACT-only, or ACT with intratumoural costimulation using a CD137 targeted antibody. Model dynamics and best fitting parameters were compared, in order to assess the mode of action of the CTLs and examine how the CD137 antibody influenced their activities. We found that the cytolytic activity of the transferred CTLs was minimal without CD137 costimulation, and that the CD137 targeted antibody did not enhance the per-capita killing ability of the transferred CTLs. Instead, the results of our modelling study suggest that an antiproliferative effect of CTLs exerted upon the tumour likely accounted for the majority of the reduction in tumour growth after CTL transfer. Moreover, we found that CD137 most likely improved tumour control via enhancement of this antiproliferative effect, as well as prolonging the period in which CTLs were inside the tumour, leading to a sustained duration of their antitumour effects following CD137 stimulation.

## 1. Introduction

The global immuno-oncology pipeline grew by 67% between 2017–2018 [1]. A substantial component of this growth came from “cell therapies”, defined in the context of immuno-oncology as therapies “that engineer immune cells such as T cells to directly attack cancer cells” [2]. Another significant component of the pipeline were therapies classified as “immunomodulators”, defined as therapies which “act on inhibitory or activating molecules expressed by T cells (…) other immune cells or the tumour immune microenvironment to unleash antitumour immunity” [2]. The rapid growth in this field reflects increasing progress in our ability to engineer Cytotoxic T Lymphocytes (CTLs) with ability to recognise and attack tumour cells, then modulate that response via therapeutic targeting of various “checkpoints”, such as the inhibitory CTLA-4 or PD-1/PD-L1 signalling axes which are among the best known immunomodulators and have been most successful in the clinic so far [3,4,5]. Although rapid growth is an indicator that the field of immuno-oncology is promising, due to the burgeoning body of literature it can be difficult to achieve consensus. In that context, mathematical and computational models are a useful tool to aid reuse and integration of previous studies. Such models can be used to integrate data from multiple sources, check their consistency, and identify those mechanisms which are most important for explaining the overall dynamics of the studied system. This systems biology approach can create novel insights into biological phenomena [6,7,8].

One immunomodulator which has received much recent attention is CD137: in 2017 there were nine trials with therapies targeting CD137; in 2018 there were 36 [2]. Although several anti-CD137 agonists are under clinical evaluation [9], the mechanisms through which anti-CD137 influences cancer immunotherapy remain under debate [10,11]. CD137 is a costimulatory molecule classified as a member of the Tumour Necrosis Factor Receptor superfamily, which is expressed on both innate and adaptive immune cells. Targeting the CD137 signalling domain has been linked to a gain of effector functions in CD8^+^ T cells including enhanced proliferation and resistance to apoptosis [12,13,14]. A recent study examined CTL functions in a B16F10 OVA expressing mouse melanoma, in combination with anti-CD137 agonist antibody administered intraperitoneally [13]. OT1 CTLs were adoptively transferred to tumour bearing mice in both the presence and absence of anti-CD137, after which tumour volume progression was recorded and the activities of tumour infiltrating CTLs were observed by means of intravital two-photon microscopy. In that study, anti-CD137 treated mice differed from control mice in the following respects: (1) Tumour bearing mice treated with adoptively transferred CTLs plus anti-CD137 (ACT+mAb) showed improved tumour control compared to counterparts treated with CTL but no antibody (ACT-only). (2) Flow cytometry analysis of cell suspensions retrieved from tumours revealed that CTLs from ACT+mAb tumours expressed greater levels of markers associated with the effector phenotype than did those recovered from tumours treated with ACT-only. (3) Intravital images gave evidence for altered CTL functioning in vivo after anti-CD137 treatment. Mitosis and apoptosis rates of both tumour and CTLs were affected, as was the migration behaviour of the CTLs. Taken together these findings support the idea that anti-CD137 together with adoptive CTL transfer improves the outcome of ACT. The suggestion was also that anti-CD137 treatment boosts the effector functions of CTLs in vivo, since anti-CD137 treatment both increased apoptosis and reduced proliferation of tumour cells compared to control. However, there was no quantification of the various effects of anti-CD137 upon CTL-mediated tumour control, so it remains unclear which enhancements to CTL effector function played the most substantial role in the improved tumour control after anti-CD137 treatment.

We previously quantified the effect of CTLs on solid tumours, considering both cytotoxicity through direct cellular interactions [6,15] and production of cytokines which inhibit tumour cell proliferation (e.g., IFNγ) as potential agents by which CTLs could control tumours [6]. We here provide a re-assessment of the in vivo melanoma data summarised above, aided by computational models. The goal was to develop a quantitative picture of the functioning of adoptively transferred CTLs in vivo, and of the effect that costimulatory anti-CD137 treatment had upon their functioning. Specifically, we aimed to determine the extent to which an antiproliferative effect contributed to tumour control, compared to direct cytotoxicity. We also asked whether the improvement in tumour control after anti-CD137 treatment was due to a numeric increase in CTLs, or due to a difference in CTL performance. To answer these questions, we developed an ordinary differential equation (ODE) model to describe the dynamical evolution of CTL treated tumours. We fit the ODE model to the in-vivo data and examined how the model parameters differed in the presence or absence of anti-CD137. We did not find any evidence that the ability of CTLs to kill tumour cells was improved in the ACT+mAb group relative to ACT only, when killing was considered on a per-capita basis. Moreover, in both ACT-only and ACT+mAb conditions, an antiproliferative effect associated with transferred CTLs explained a far larger share of the reduction in tumour progression than did CTL cytotoxicity towards tumour cells. Finally, an increased antiproliferative effect associated with anti-CD137 treatment, together with a more sustained presence of CTLs within the tumour after anti-CD137 treatment, could explain the reduced tumour progression in our data.

## 2. Materials and Methods

### 2.1. Experimental Data

The experimental data we have used derives from a previously published study where mice were inoculated with B16F10 melanoma tumours, which were then studied over a period of up to 15 days post-inoculation through dorsal imaging windows, by means of two-photon microscopy [13]. In total there were 6 distinct experimental conditions. There were two control conditions, one with OVA antigen expressing tumour cells where ACT was not administered, and another with tumour cells not expressing OVA, but with ACT administered. A further two conditions had OVA expressing tumour cells, with ACT administered on either day 3 or day 7 post tumour inoculation. Finally, there were two conditions where ACT was administered to mice bearing OVA expressing tumours, together with costimulation using agonist anti-CD137 (clone 1D8), again on either day 3 or day 7 post tumour inoculation. For these costimulated conditions, anti-CD137 was delivered intraperitoneally on the same day as ACT.

The dataset comprises estimates of tumour volume measured at days 1, 3, 6, 9, 13 and 15 in all mice. Moreover, for the mice treated with ACT on day 3 and bearing OVA expressing tumours, statistics were available for the number of CTLs and tumour cells, as well as their mitosis and apoptosis rates, on either day 6 or day 9 after tumour inoculation (days 3 and 6 after CTL transfer). These statistics are samples from imaging volumes of size 0.35 × 0.35 × 0.1 mm which were imaged for 1–3 h (See Figure 6B,D of ref: [13]). Finally, intravital images were available for the mice treated on day 7, although no statistics for these images had yet been recorded. We therefore quantified the number of CTLs and tumour cells, and the number of tumour cell mitosis events in these images as well. The number of CTLs and TC mitosis events was determined via manual counting, whereas the number of tumour cells was determined automatically using ImageJ as follows: we first processed images using a 3D gaussian blur (sigma = 2 in the x-y directions, and sigma = 0.2 in the z direction). Then, we selected every third slice in the z direction to avoid repeated counting of the same cell. Remaining slices were then subjected to a threshold using the Li method [16]. Following thresholding, the watershed method [17] was used to separate touching cells. Finally, the “analyse particles” plugin was used to quantify cells, excluding particles of less than 10 pixels in area.

To determine the killing rate from the experimental data we considered the number of apoptosis events counted per position as a Poisson distributed random variable. We evaluated two possibilities for the Poisson rate parameter, which we termed “linear” or “mass-action”. For the linear model, we considered the rate parameter to be proportional to the number of CTLs counted at the position. For the mass-action model, we considered the rate parameter to be proportional to the product of the number of CTLs and the number of tumour cells counted at the position. For either model, we determined the rate parameter which maximised the likelihood of the counted apoptosis events from the Poisson distribution. We considered all samples together, or alternatively samples grouped by treatment, resulting in different numbers of fitted rate parameters required to describe the data. To compare the quality of fits while accounting for different numbers of parameters, we used Akaike’s and Bayes’ information criteria.

### 2.2. Ordinary Differential Equation Model CTLs vs. B16F10 Melanoma

We used ODE models for tumour growth, CTL population dynamics and the effect of the CTLs on tumours, thus approximating tumours as well-mixed entities. Although some degree of intratumoural heterogeneity can be expected, considering them to be spatially homogeneous is a reasonable initial approach given the small size of the tumours. Several ODE models have previously been proposed to describe the tumour-immune interaction [18,19,20,21], although these were not developed with the benefit of detailed intravital measurements concerning cellular apoptosis and mitosis rates such as those we employ here. Therefore, whilst we employ elements from previously published models, we also include some modifications which allow us to achieve a satisfactory match to the available experimental data.

#### 2.2.1. Tumour Growth in the Absence of CTLs

The available tumour growth data derives from B16F10 murine melanoma tumours recorded for 15 days after their implantation [13]. We opted for the simplest possible model for tumour growth in the absence of CTLs, i.e., the exponential growth model which considers a volume of tumour (V) made of tumour cells who undergo mitosis with an average rate g, because it was sufficient to describe the part of the data without ACT very well. Indeed, within the 15-day time period we studied, tumours remained small: the median tumour volume (considering all our data) was 0.04 mm^3^ on day 3 and 0.93 mm^3^ on day 15. Thus, there was no need to take into account a reduction in TC proliferation as the tumour volume increases, as is commonly seen when models are applied to large malignancies and the logistic growth model is applied [18,19,20,22]. For all our simulations we took g = 0.5 (day^−1^) in the absence of CTLs, based on growth rates for tumours in the data for conditions either without ACT or periods before CTL transfer in the ACT treated groups. This corresponds to a doubling time of 1.4 days for the tumour cell population, and it is approximately consistent with doubling times in the region 1.7–2 (day^−1^) reported for B16F10 tumours growing up to 100 mm^3^ in another study, where B16F10 cells were implanted into the ears of mice [23] (instead of the deep dermis as was the case for the data we studied here).

#### 2.2.2. Effects of CTLs on Tumours

We consider the following model for the killing rate of tumour cells by CTLs:(1)$$k_{DS}=\frac{kET}{1+E/h_{e}+T/h_{t}},$$
which is known as the dual saturation model [24]. Similar to the commonly used mass-action model of CTL killing [19,25,26], the dual saturation killing rate presented in Equation (1), $k_{DS}$, increases linearly with the number of CTLs, *E*, and the number of tumour cells, *T*, so the parameter k is analogous to a mass-action killing rate. Differently to the mass-action model, the dual saturation model allows for saturation in the rate of killing when the number of either CTLs or tumour cells becomes large. The saturation constant $h_{e}$ determines the extent of saturation in killing for CTLs, whilst $h_{t}$ determines the extent of saturation in killing for the number of tumour cells. Since the tumours we deal with here are densely packed with tumour cells and the frequency of infiltrating CTLs is relatively low, we make the simplification that $1+E/h_{e}\ll T/h_{t}$, such that Equation (1) reduces to $kh_{t}E$ and CTLs kill at a constant rate $k_{e}=kh_{t}$ (CTL^−1^ day^−1^). An advantage of this simplification is that the killing rate $k_{e}$ can be determined using the limited amount of available data, whereas these data would not allow for determination of the parameters k, $h_{e}$ and $h_{T}.$ A disadvantage is that the resulting model remains valid only in regimes where tumour cells are frequent and the density of infiltrating CTLs is relatively low. The value of the parameter $k_{e}$ should be determined by factors such as the antigenicity of the tumours and the susceptibility of the tumours to CTL mediated cytotoxicity, which we take to remain constant over the duration of the experiments described here. We found no evidence in the data to suggest that killing of tumour cells by CTLs might depend on the number of tumour cells (Appendix A).

In our data the frequency of tumour cell apoptosis was low, so in addition to the killing we included an antiproliferative effect of CTLs on the tumour in our models. For this, we denote separately $T_{p}$, the subset of tumour cells which are proliferating and $T_{q}$, a subset of “quiescent”, non-proliferating tumour cells:(2)$$T_{p}\;+\;T_{q}\;=\;T,$$
where *T* in Equation (2) is the total number of tumour cells in the tumour. Our quiescent state is motivated by observations of an IFNγ-dependent cell-cycle arrest in B16F10 melanoma after ACT [27], which we previously implicated in control of murine EL4 lymphoma [6] and for which there is also evidence in ovarian and breast carcinoma models [28]. In our model, induction of the quiescent state happens at rate $k_{q}$ (CTL^−1^ day^−1^). CTLs secrete IFNγ upon encounter with antigen presenting cells [29], therefore the value of $k_{q}$ will be determined by factors such as the antigenicity of the tumour as well as the susceptibility of the tumour cells to IFNγ-dependent cell-cycle arrest. In the absence of cognate antigen, no cell-cycle arrest is expected since CTLs would not produce IFNγ. In B16F10 melanoma, quiescent tumour cells recover from CTL induced cell cycle arrest after a few days [27]. In our model recovery occurs with rate $d_{q}$ (day^−1^). The dynamics of proliferating and quiescent tumour cells can thus be described as in Equations (3) and (4):(3)$$dT_{p}/dt\;=\;gT_{p}\;-\;\left(k_{e}\;+\;k_{q}\right)\left(T_{p}/T\right)E\;+\;d_{q}T_{q},$$
(4)$$dT_{p}/dt\;=\;k_{q}\left(T_{p}/T\right)E\;-\;k_{e}\left(T_{q}/T\right)E\;-\;d_{q}T_{q}.$$

Note that the killing rate is shared between proliferating and quiescent tumour cells in accordance with their fraction in the tumour.

#### 2.2.3. CTL Population Dynamics

Our model considers only transferred CTLs (since only these could be seen with the fluorescent reporter system), therefore neglecting any endogenous response. Although endogenous CTLs may have been present, robust tumour control in the presence of ACT was still achieved in RAG −/− mice (See Figure 1H in reference [13]) for approximately 30 days before tumours regrew. From this we concluded that endogenous CTLs did not contribute significantly to early tumour regression in the experimental set-up we model here but were required for long term tumour control. Negligible contribution of the endogenous (or innate) immune response are also consistent with our observation that TC apoptosis correlated strongly with the local number of transferred CTLs (See Appendix A), although we cannot exclude that the local density of endogenous/innate effectors might also have correlated with the local density of transferred CTLs. The total number of tumour-infiltrating CTLs, E, inside the tumour is described by Equation (5):(5)$$dE/dt\;=\;sT^{2/3}\;+\;E\left(I-R\right).$$

The first term represents the net movement of transferred CTLs into the tumour. Since we have no measurements directly pertaining to CTL infiltration of the tumours, we opted for a simple model where the net movement of CTLs across the tumour boundary occurs at a constant rate s (per unit area of the boundary). Parameter s is expected to depend on the ability of transferred CTLs to recognize the tumour, thus should depend on the antigenicity of the tumour. The 2/3 power can be interpreted as a constant rate of infiltration across the boundary of the tumour, which we consider to be approximately spherical [30]. We consider this spherical approximation valid despite the presence of the imaging window, since the mice were sacrificed whilst the tumours remained small (<10 mm^3^), therefore any distortion in tumour shape due to the presence of the imaging windows should be minimal. To take the CTL population dynamics into account, the second term of Equation (5) includes two additional variables *I* and *R* to describe the dynamics of CTL proliferation and apoptosis inside the tumour. The variable *I* represents an auto-inductive response of CTLs upon encounter with antigen expressing cells and is based on other models which have included Interleukin-2 as a driver of CTL mitosis [18,21]. Such stimulatory signals could originate from other CD8^+^ T cells by means of quorum regulation [31] or from other immune cells such as CD4^+^ T cells. Our model accounts for either possibility:(6)$$dI/dt\;=\;k_{I}E/T-d_{I}I.$$

Thus, CTLs induce their own mitosis at rate $k_{I}$ and the stimulus disappears at rate $d_{I}$. Differently to a previous model [18] where the stimulatory term was proportional to the number of tumour cells, in our model the production term for I (i.e., the first term in Equation (6)) is proportional to the *E*:*T* ratio, as the frequency of CTLs among tumour cells should determine the strength of the auto-inductive mechanisms.

The second variable *R* represents a resistance acquired by the tumour in response to infiltrating CTLs, leading to an increase in the rate of CTL apoptosis over time. This variable rate of CTL apoptosis represents a difference between our model and previous models in the literature which assume a constant apoptosis rate for CTLs [18,21,25]. Including this change was necessary since in our case data for the apoptosis rate of CTLs was available, and this rate clearly varied over time. Pro-apoptotic signals through the PD-1 receptor are a candidate source of this resistance, since in other experiments with B16F10 tumours treated with agonist antibody for CD137 tumour rejection was enhanced when agonist anti-CD137 was coadministered with an antagonist antibody for PD1 [32]. However, since other possible explanations are equally consistent with our data (see Section 4), we used the general term “resistance” for this variable:(7)$$dR/dt\;=\;k_{R}E/T\;-\;d_{R}R.$$

Similar to the auto-inductive effect of CTLs, in Equation (7) CTLs induce resistance proportional to the *E*:*T* ratio inside the tumour at a constant rate $k_{R}$, with this resistance decaying over time with rate $d_{R}$. During our parameter estimation we frequently found very low values for the parameter $d_{R}$. This was likely due to the short timescale of the experiments, implying there was not sufficient time for this resistance to decrease substantially. Therefore, for simplicity we fixed this parameter to $d_{R}=0$. Note that since we do not have any direct data regarding the underlying mechanisms causing the auto-inductive or resistive phenomena, for simplicity we have expressed the variables *I* and *R* in terms of their effects on the mitosis or apoptosis rates of CTLs. Thus, *I* and *R* have units of events CTL^−1^ day^−1^ and additional scaling parameters in the second term of Equation (5) are unnecessary.

#### 2.2.4. Model Initialisation

Models were initialized on day 0, corresponding to the day of tumour inoculation in the mice. The initial conditions were *T* = 1200, *E* = 0, *I* = 0, *R* = 0. Since our model only describes the adoptively transferred CTLs, we set the infiltration parameter *s* = 0 until the day of CTL transfer (either day 3 or day 7). On the day of CTL transfer, the parameter *s* was immediately set to the best fitting value thus allowing CTLs to begin infiltrating the tumours.

#### 2.2.5. Model Fitting Procedure

Models are fit by minimising the Root Mean Square Error between model prediction and all individual data points. “Individual data points” are considered to be either one volumetric growth estimate, or one statistic estimated from one intravital position. Thus, a single mouse where the tumour volume was measured on days 1, 3, 6, 9, 13 and 15 and where intravital data was recorded at four positions on days 6 and 9 would produce 5 volumetric growth estimates, plus 8 estimates for each intravital process rate (32 in total) and a further 8 E:T ratio estimates. The correspondence between the intravital process rates determined from the experimental dataset and those determined from the ODE model is given in Table 1. ACT-only and ACT + mAb groups were fit separately with no overlapping parameters, except for the tumour growth rate parameter (*g*) which was fixed to the same value for both ACT-only and ACT + mAb conditions before fitting. For each group we varied all the parameters to simultaneously minimise the errors for all available data points, for both day 3 and day 7 treated conditions together. This simultaneous fitting improves identifiability of the model by supplying additional constraints. For example, the intravital process dynamics are constrained by the requirement for the model to also describe the tumour volume progression (for which more time points are available). The intravital dynamics within the day 7 treated conditions (for which few measurements were available) are also constrained by the requirement for the model to describe the intravital dynamics within the day 3 treated tumours. Minimisation was performed using a differential evolution algorithm with the DEoptim [33] package in R, using the local-to-best evolution strategy. In brief, the differential evolution algorithm is an optimization procedure inspired by natural selection, in which an initial population of parameter vectors is refined over successive generations until the parameter vector which optimizes the objective function is found. New parameter vectors (children) are formed from a previous generation by combining and perturbing parameter vectors from the previous generation (parents). Both child and parent parameter vectors are evaluated and the parameter vector which results in the lowest value of the objective function proceeds to the next generation. During the optimization procedure we searched within the bounds 0–50 (day^−1^) for parameters k_e_, k_i_, k_q_, k_r_, d_i_, and d_r_. For the parameter s we searched within bounds of 0–5 (TC^−2/3^ day^−1^). Each parameter estimation attempt was performed with 5 repeats, using different randomly selected starting parameter values for each repeat. Individual repeats had population sizes of 200 and ran for 500 generations.

## 3. Results

### 3.1. Tumour Cell Apoptosis Only Exceeds Mitosis in Mice Treated with CD137 Antibody

In this work our aim was to integrate dynamic two-photon imaging and volumetric tumour progression data [13], to create a systems-based description of a murine melanoma after ACT. To understand the expansion and retraction of CTL populations in the tumour during therapy and in relation to local tumour response, we first plotted apoptosis against mitosis rates of tumour cells (Figure 1A) and of CTLs (Figure 1B), both stratified by mouse (Figure 1A,B, shapes) and by the day of measurement (Figure 1A,B, colours). Apoptosis and mitosis events were derived from 2 h time-lapse sequences, allowing accurate calculation of net cell proliferation or regression rates. Each plot splits into two regions: net population growth when mitosis exceeded apoptosis (Figure 1A,B, above dashed lines) and net population reduction otherwise (Figure 1A,B, below dashed lines). There was net growth of tumour cells in all except one measured position with ACT only (Figure 1A, left panel), whereas with ACT + mAb this was true for only half of the measured positions (Figure 1A, right panel). In contrast to tumour cells which were mostly proliferating, CTL apoptosis matched or exceeded mitosis in almost all measured positions (Figure 1B), suggesting that transferred CTL populations were only able to sustain their numbers, but were not “expanding” inside the tumour. Nevertheless, we observed much higher absolute rates of apoptosis or mitosis for CTLs compared to tumour cells (compare axes values between Figure 1A,B), suggesting that there is more potential for rapid changes in the number of CTLs compared to tumour cells inside the tumour if CTL apoptosis could be reduced relative to mitosis.

Besides mitosis and apoptosis rates (Figure 1A,B), the intravital dataset also consists of the total numbers of each cell type which are represented by point size in separate plots for tumour cells (Figure 1C) or CTLs (Figure 1D). Considering the data per-mouse (Figure 1A,B, shapes or Figure 1C,D, colours), it is apparent that all measurements from the same day pertaining to a given tumour are clustered (valid for both CTLs and tumour cells). Similar measurements at different sites within one tumour indicated that tumours were spatially relatively homogeneous, at least for the peripheral areas that were imaged in the study. However, the fact that the clusters travel over time indicates that conditions inside the tumours were not temporally homogeneous.

The number of either CTLs or TCs per position was similar between ACT-only and ACT + mAb groups, although CTL numbers were slightly higher in ACT + mAb conditions (Figure 1C,D, comparing within figures). TCs generally outnumbered CTLs in all positions (Figure 1C,D, comparing between figures). Comparing tumour cells between ACT-only and ACT + mAb (Figure 1A,C; columns), the population dynamics appear most different in two of the mice, both corresponding to the ACT + mAb group (m65 d6&d9, m49 d6). Measurements from those mice occur in the region of the plot below the red dashed line where local tumour regression is apparent, which was barely reached in any ACT-only mouse. Interestingly, the mitosis rates of tumour cells in the ACT-only group decrease between days 6–9, yet over the same time interval TC mitosis rates in the ACT+mAb group increase (Figure 1E). This result suggests that tumours were recovering proliferative capacity after ACT + mAb treatment faster than the ACT-only treated group, seemingly at odds with the more sustained CTL activity previously reported [13]. Similarly, there was a more sustained replacement of the CTL population in the ACT + mAb group compared to the ACT-only group (Figure 1B,F, solid lines in right panels remain parallel to the dashed line), which was due mainly to an increase in mitosis rather than to a decrease in apoptosis. Overall, these results indicate that the ACT + mAb treatment improved tumour control by shifting tumour cell dynamics towards a regime where net apoptosis exceeded mitosis, but it is unclear from this analysis how CTLs participated in this process.

### 3.2. The Presence of CTLs Leads to Increased Tumour Cell Apoptosis and Decreased Tumour Cell Mitosis

In addition to the intravital statistics, volume progression data for the tumours from the same experiments were also available (Figure 2A). There were two control conditions: one where ACT was applied on day 3 but tumour cells did not express the cognate OVA antigen for recognition by adoptively transferred CTLs (Figure 2A, top row left), and another where tumour cells expressed OVA but mice did not receive ACT (Figure 2A, top row right). In the middle row are the two experiments (corresponding to Figure 1 intravital data) where ACT was applied 3 days after mice were inoculated with tumours, and in the final row ACT treatment was delayed until day 7 after tumour inoculation. Data points from mice that had not (yet) received ACT are black, whereas points from mice that had received ACT are green. To compare these volumetric data with the intravital dataset we converted the volumetric data for each mouse into growth rates (Figure 2B), i.e., each point represents the growth rate of a single tumour between successive volume measurements. This conversion ensured that all our data points later used for model fitting would have the same units (day^−1^). We estimated that the growth rate of the untreated tumours was approximately 0.5 day^−1^ (Figure 2B, Appendix A grey lines; also slope of grey lines in Figure 2A). The impact of the transferred CTLs is clear from the transient decrease in volumetric growth rate observable after ACT (Figure 2A,B, green lines 2nd and 3rd rows).

To verify whether the activity of the CTLs observed in the intravital data was consistent with the measurements of tumour progression based on the volumetric data, we considered two possible effects of CTLs on tumours: either killing of tumour cells by CTLs, or prevention of proliferation. In the intravital dataset we studied whether the number of tumour cells per position influenced the killing rate of CTLs but found no clear evidence that this was the case (Appendix A). We found that the number of TC apoptosis events per position could be adequately described by a straightforward Poisson model [34], with the intensity of the killing directly proportional to the number of CTLs (Figure 2C left panel). The correlation between TC mitosis and CTL numbers was less clear, although appeared to be negative since the positions with the most tumour cell mitosis were those with few CTLs (Figure 2C right panel, Appendix A). Thus, the intravital data suggest the presence of CTLs led to killing of tumour cells and inhibition of their proliferation, which is consistent with the volumetric data. However, it was unclear which of these effects were most important in the control of the tumours.

### 3.3. Ordinary Differential Equation Model Describes CTLs vs. B16F10 Melanoma

In order to probe the relative contributions of these two effects (antimitotic or killing) we elected to develop an ODE model to combine all the disparate measurements together and check them for internal consistency and with other reports in the literature. The absolute number of tumour cells per field in the imaging data depended strongly on the location of the imaging windows which were sometimes located in the centre of the tumour but other times near the periphery or contained large features like vessels (Figure 3A). Therefore we opted to discard the absolute numbers of CTLs and tumour cells, instead using the CTL:TC (Effector:Target, *E*:*T*) ratio to develop our ODE model. A further advantage of utilising the dimensionless *E*:*T* ratio for fitting is that it prevents the physical size of the imaging windows unduly influencing our results. Our ODE model (Figure 3B) features CTLs (*E*) either killing tumour cells (*T*) or preventing them from proliferating, which in our model happens via transfer of proliferating (*T_p_*) tumour cells into a quiescent state (*T_q_*). To describe the population dynamics of CTLs we considered CTLs to infiltrate across the tumour boundary at a constant rate *s* per unit area of boundary. Additionally, since CTL mitosis and apoptosis measurements within the tumour were available we included these processes in our model as well via incorporation of two loops. The first loop considered a factor stimulating CTL proliferation (Induction/Interleukin: *I*), whereas the second loop described a tumour resistance factor (*R*). The resistance factor increased over time spent with CTLs inside the tumour and led to an increase in CTL apoptosis (see Section 2). We fit our model simultaneously to all available measurements, except for the tumour growth rate in the untreated condition, which we fixed before fitting (using g = 0.5 day^−1^). Overall we varied 7 parameters for either the ACT-only or the ACT+mAb condition (Figure 3C and Table 2) to obtain the best match between model and tumour data. Since the model was fit to the tumour growth rate data rather than the tumour volume progression data, we verified whether the best fitting parameters would match to the tumour volume progression data (Appendix A). Indeed, this resulted in a generally good match, although slight mismatches were observed for late time points of tumours treated on day 3. A local sensitivity analysis within a narrow range around the best fitting parameter sets indicated that a small change to most parameters would compromise the ability of the model to describe the experimental data (Appendix A). In conclusion, the ODE model that we developed is able to describe the experimental data.

### 3.4. An Antiproliferative Effect of CTLs Is Most Important for Controlling Tumour Progression

We studied our best fitting models to gain insight into the dynamics and activities of the transferred CTLs. When we took best fitting parameter sets for either ACT-only or ACT+mAb conditions and varied the killing rate k_e_, abrogation of killing (by setting k_e_ = 0; Figure 4A) had only marginal impact on the progression of the tumours. In contrast, with abrogation of CTL-induced tumour proliferation arrest (by setting k_q_ = 0; Figure 4B) tumour growth progression continued virtually unaffected by the presence of the CTLs killing at the best fitted rates. This indicated that the antiproliferative effect, rather than the killing, accounted for the majority of deviation from exponential growth. Notably, when we simulated progressive increases to CTL killing, we saw progressively improved tumour control for both ACT-only and ACT+mAb conditions (k_e_ > 1; Figure 4A). However, the antimitotic effect appeared to be close to saturation, particularly for the ACT+mAb condition, since further increases to the k_q_ parameter hardly led to further improvements in tumour control (k_q_ > 1; Figure 4B). Note that these results may overestimate the importance of killing for large values of k_e_ at late time points where the tumor has substantially regressed, which is due to the simplified killing term we employed (Section 2.2.2). Nevertheless, our analysis does imply that strategies for increasing CTL killing (k_e_) could be of greater therapeutic benefit than strategies aiming to further enhance the antimitotic effect (k_q_).

### 3.5. Anti-CD137 Leads to Superior Tumour Control by Enhancing the Antiproliferative Effect of CTLs

We finally sought to identify differences in the dynamics of CTLs and their interactions with the tumour after the CD137 antibody costimulation. Since abrogating killing by setting k_e_ = 0 had no substantial impact on tumour volume progression (Figure 4A), it seemed unlikely that ACT+mAb enhanced control of the tumour by improving the ability of CTLs to kill tumour cells. Indeed, there was no improvement in the per-capita CTL killing performance of CTLs after ACT+mAb. In fact the best fitting killing rates from ACT-only (k_e_ = 0.75 day^−1^) were even somewhat higher than for the ACT+mAb condition (k_e_ = 0.5 day^−1^) (Figure 3B and Figure 5A,B, row 2).

Since the E:T ratio was also higher in the ACT+mAb tumours (Figure 5C,D), increased killing due to more CTLs might explain the improved tumour control in the ACT+mAb group. There are multiple indications that this was not the case. First, tumour growth reduction was broadly similar between tumours treated with ACT+mAb on d3 (Figure 5E) and those treated on d7 (Figure 5F), despite substantially lower E:T ratios in the d7 treated group—consistent with the notion of a small number of CTLs being quickly able to control a large number of tumour cells through cytostatic effects. Second, the tumour cell mitosis predicted by the best fitting models followed the volumetric tumour growth closely, leaving little room for a contribution from killing (compare Figure 5E,F with Figure 5A,B, top row).

Rather than an increase in killing, a stronger reduction in tumour proliferation for ACT+mAb tumours compared to ACT-only tumours accounts for the difference in results. Note that different values for the preset growth rate parameter resulted in similar best fitting parameters and similar model dynamics, therefore our conclusions about the relative importance of killing versus antiproliferative effect do not appear to be especially sensitive to our choice for the tumour growth rate (Appendix A). Our model suggests two means by which enhanced reduction of proliferation for ACT+mAb tumours could have occurred. First, our fits resulted in larger values of k_q_ for the ACT+mAb group (Figure 3C), which is necessary to account for the similar reductions in volumetric growth in the ACT+mAb treated tumours whether treated on d3 or d7 (Figure 2B, blue lines), despite a much lower *E*:*T* ratio in the d7 treated group. The second possibility our model highlights stems from the different dynamics of the CTL population between ACT-only and ACT+mAb treated groups. At late times the population dynamics of the CTLs was an important determinant of the E:T ratio, due to a combination of slow dynamics for CTL mitosis (Figure 5A,B, row 3) and a delayed onset of CTL apoptosis (Figure 5A,B, row 4). Notably, although the fitted rate parameter controlling the increase in CTL mitosis (k_i_) was substantially larger for the ACT-only condition than for the ACT+mAb condition (Figure 3C), this did not reduce CTL mitosis overall—instead, the peak of CTL mitosis simply shifted to later time points. These altered dynamics led to significant improvement in *E*:*T* ratio for the ACT+mAb group at late time points after treatment. Overall, our model indicates that ACT+mAb costimulation resulted in CTLs which were able to more rapidly prevent tumour cell mitosis after administration of ACT. Moreover, the dynamics of both CTL proliferation and apoptosis were delayed, resulting in a CTL population which remained inside the tumours for longer and therefore increasing the period of time in which CTLs could exert control of the tumour.

## 4. Discussion

Here we used an ODE model to quantify the effector functions and population dynamics of CTLs and tumour cells, following ACT therapy within murine melanoma tumours. To parameterise our models we used data where mice were treated in the presence or absence of anti-CD137 [13]. The data consisted of counts of the number of CTLs/tumour cells and the number of apoptosis/mitosis events associated with each respective cell type, at various locations within the melanoma tumours being attacked by CTLs. We used our models to investigate the means by which adoptively transferred CTLs controlled the tumours, and also what caused the improved tumour control after anti-CD137 costimulation. We found that the apoptosis rates of tumour cells were well fitted by a linear dependency on the number of CTLs, indicating that local presence of CTLs was required for TC apoptosis. However, we found that the CTL killing rate was very low and contributed little to tumour reduction overall. Instead, the antiproliferative effect had a large effect, with tumour cell mitosis rates observed in vivo being far below those needed to explain the growth of the untreated tumours. We found that almost all of the reduction in tumour growth after CTL treatment could be explained by decreased TC mitosis.

Importantly, we also sought to understand mechanistically what caused the improved tumour control after treatment with anti-CD137. Killing per CTL was unchanged after anti-CD137 stimulation and hence still contributed little to tumour control, so the improvement in tumour control was largely due to an enhanced antimitotic effect after anti-CD137 treatment. Our model suggested that this enhanced antimitotic effect could be explained either by an increased per-capita ability of CTLs to exert an antimitotic effect, or simply by a numeric increase of CTLs inside the tumour (each with similar antimitotic effects to unstimulated CTLs when considered per-capita). In our model these effects were difficult to separate, since increased mitotic effect of CTLs (on a per-capita basis) should also have the effect of increasing *E*:*T* ratio, by reducing the denominator. However, differences in *E*:*T* ratio between ACT-only and ACT+mAb treated groups emerged later than the differences in tumour cell proliferation and tumour size, which were already apparent by day 6 after tumour inoculation. Thus, the data from early time points suggest a more rapid per-capita ability of anti-CD137 stimulated CTLs to prevent proliferation of tumour cells, compared to their unstimulated counterparts. Additionally, our model predicted that the increased *E*:*T* ratio, which was most apparent at late time points after ACT, should also play a role in improved tumour control. Although the rates of CTL mitosis were generally low and the net CTL population growth (mitosis minus apoptosis) was negative in almost all videos, the overall dynamics of the anti-CD137 stimulated CTL populations appeared different to their unstimulated counterparts. Specifically, the peak rate of CTL mitosis occurred later in the ACT+mAb treated group, which together with a delayed onset of apoptosis led to a more sustained presence in the tumour. In summary, after anti-CD137 treatment CTLs were able to rapidly shut down mitosis of tumour cells, but also remained present in the tumour for longer, both contributing to the improvement in tumour control.

Our results fit well with other reports about the effects of anti-CD137 stimulatory effects on CTL function in the literature. To our knowledge, there are no clear reports that CD137 enhances the cytotoxicity of CTLs. Instead, in agreement with our findings, studies which have directly measured CTL cytolytic activity have found similar killing after blockade of CD137 signalling [35], and CAR T cells engineered with a CD137 costimulatory module did not exhibit superior cytotoxic potential compared to CAR T cells lacking the CD137 module [36]. Several studies have found that CD137 costimulation induces IFN-γ production by CTLs [35,36,37]. Enhanced IFN-γ production by CTLs provides a possible mechanism for the increased ability of CTLs to prevent tumour cell proliferation suggested by our models, since IFN-γ has been shown to play an important role in control of B16F10 melanoma tumours via arrest of the tumour cell cycle [27]. The prolonged presence of CTLs inside the tumour for the anti-CD137 treated group due to delayed apoptosis is in agreement with reports of anti-apoptotic effects of CD137 signalling on activated T cells [38,39].

One limitation of our model is that there is no representation of space, so the tumour is treated as homogeneous throughout. The B16F10 melanoma tumours are highly invasive [40] and events at the invading margins may be more important than events elsewhere in determining tumour growth, with tumour cells near the periphery having more space and more opportunities for proliferation. Anti-CD137 treatment reduced CTL migration inside the tumour resulting in long-lasting interactions with tumour cells [13], so it may be that CTLs remained near the tumour border or tumour vasculature and were more effective here than the control cells which migrated deeper into the tumour. Our model would not be sensitive to such an effect. Furthermore, we took the tumour cell density as constant but in reality, this may have reduced if tumour cells continued to migrate outwards but proliferation was inhibited and killing occurred. In this case our estimates of the number of tumour cells are too high and our model underestimates the impact of the antiproliferative effect on the tumour, possibly explaining the remaining error in the tumour cell mitosis rate for our best fitting models. A spatially explicit model such as a partial differential equation model [41,42] could be developed, to take into account these limitations. This would however increase the complexity of the model, so more detailed measurements from the tumour would be required in order to determine the model parameters. Specifically, measurements of mitosis and killing rates categorised based on the distance to the centre or periphery of the tumour would be useful to parameterise such a spatial model. Another useful measurement would be the net migration rate of both CTLs and tumour cells, along with the direction of migration. Moreover, another limitation of our model is the low number of temporal data points for the intravital measurements. It would be very valuable to have a more detailed time course for CTL infiltration into tumours.

Our model predictions can be tested in various ways. Although we included the tumours treated on day 7 in our analysis, we were not able to determine the number of TC apoptosis events or the apoptosis/mitosis of CTLs. That was because, whilst absolute cell numbers or TC mitosis events were relatively easy to detect, the other events were ambiguous. Nevertheless, our model does make predictions for these values which could in principle be checked. Moreover, our model predicts the arrival rate of new CTLs into the tumours which could be checked against time-lapse images. In our data CTL apoptosis increased over time, so we introduced a resistance variable R to account for this. This is consistent with recent reports that long range IFN-γ signalling can cause upregulation of PD-L1 in cells across distances of up to 800 μm [28]; or that IFN-γ-dependent invasion of myeloid-derived suppressor cells could be the major source of suppression [43]; or other immune checkpoint death receptors such as FAS-L [44]; or perhaps competition between tumour cells and CTLs for nutrients was a primary mediator of CTL apoptosis in our model, since both activated CTLs and tumour cells rely heavily on anaerobic glycolysis or glycolysis as sources of fuel [45], and interactions between CTLs and stromal cells resulted in catastrophic destruction of tumour vasculature [46], which ought to result in a reduction in glucose supply to the tumour and might account for increased apoptosis of CTLs over time. It would be useful to acquire more experimental data which could shed light on the reasons for the observed time-increasing apoptosis rate.

In future, our modelling approach could be extended to include human data. For such an extension, it is a prerequisite that the model should be defined in terms of variables and parameters that can feasibly be measured in human patients. For certain variables this is already possible, for example the frequency of tumour infiltrating CTLs can be obtained from tumour biopsies using immunofluorescence techniques [47] or from gene expression data using signature gene expression profiles [48]. Other variables such as cell mitosis rates were measured here using live intravital imaging, thus could not be obtained from human patients in the same way. Nevertheless, information about cell mitosis could be obtained from analysis of cell cycle marker genes [49]. Therefore, the next steps should be to expand our modelling approach to additional experimental tumours, while parameterizing models with measurements made using modalities that are feasible to use with human patients. Such an intermediate approach could validate our modelling approach in the absence of the detailed intravital imaging measurements we have used here.

## 5. Conclusions

Overall, our modelling study provides insights into the mechanisms CTLs use to control tumours, as well as insights into how these mechanisms may have changed upon costimulation with agonist antibody targeting the CD137 receptor. We found that the cytolytic activity of transferred CTLs against tumour cells was low, with an antiproliferative effect exerted upon tumour cells being a more important factor in tumour control. Moreover, our model suggested that a stronger antiproliferative effect together with a more sustained presence of CTLs inside the tumour were the main effects of costimulation via the CD137 receptor, which together led to improved tumour control. The results of our model identify specific directions for future experimental work which would help elucidate the effect of CD137 stimulation upon CTLs.

## Figures and Tables

**Figure 1 cancers-13-02567-f001:**
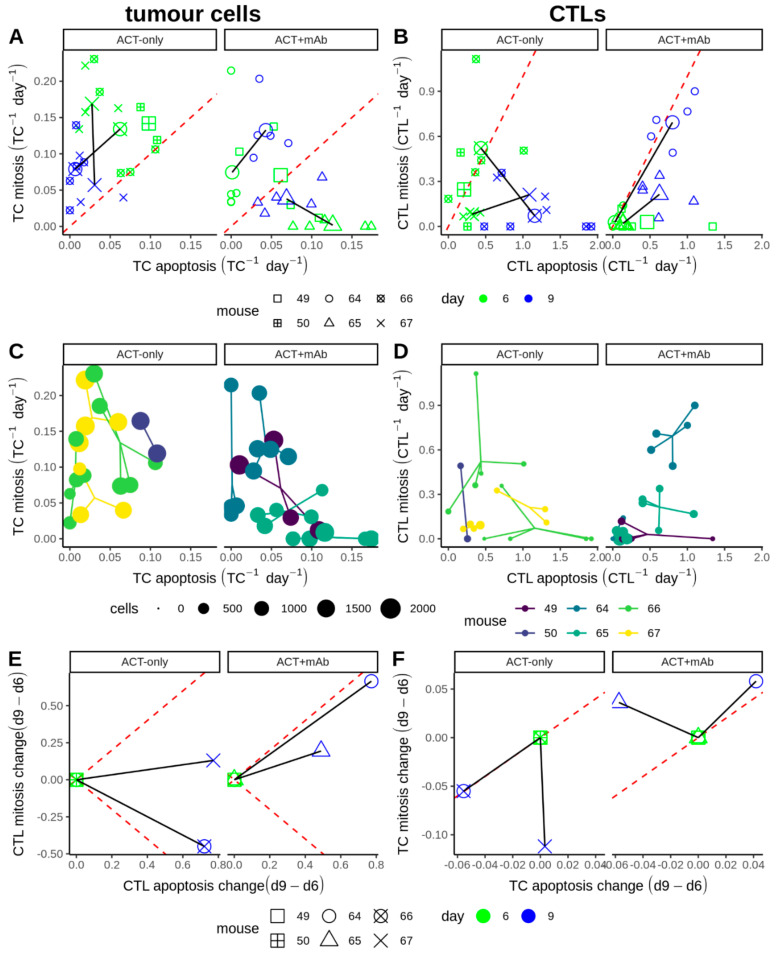
Comparison of apoptosis and mitosis rates for tumour cells and CTLs. (**A**,**B**) Apoptosis and mitosis rates of tumour cells (**A**) or CTLs (**B**) with or without anti-CD137 (columns). Each small point represents two simultaneous apoptosis (*x*-axis) and mitosis (*y*-axis) rates measured at one site within a tumour. Points are coloured based on the day of measurement, and different mice are indicated by shape. Large points are the mean values per position/day; these are connected by solid black lines for cases where we have intravital measurements on both days 6 and 9 from the same mouse. The red dashed line marks net zero population growth. (**C**,**D**) Apoptosis and mitosis rates of tumour cells (**C**) or CTLs (**D**) where point size indicates the total number of cells recorded per site. Segments connect all points from the same mouse imaged on the same day. Each colour represents a different mouse. (**E**,**F**) Change in the apoptosis and mitosis rates of tumour cells (**E**) or CTLs (**F**) based on intravital data for two mice per condition (constructed via linear translation of the mean values in **A**,**B** such that the day 6 measurement lies at the origin). Points are coloured based on the day of measurement, and different mice are indicated by shape.

**Figure 2 cancers-13-02567-f002:**
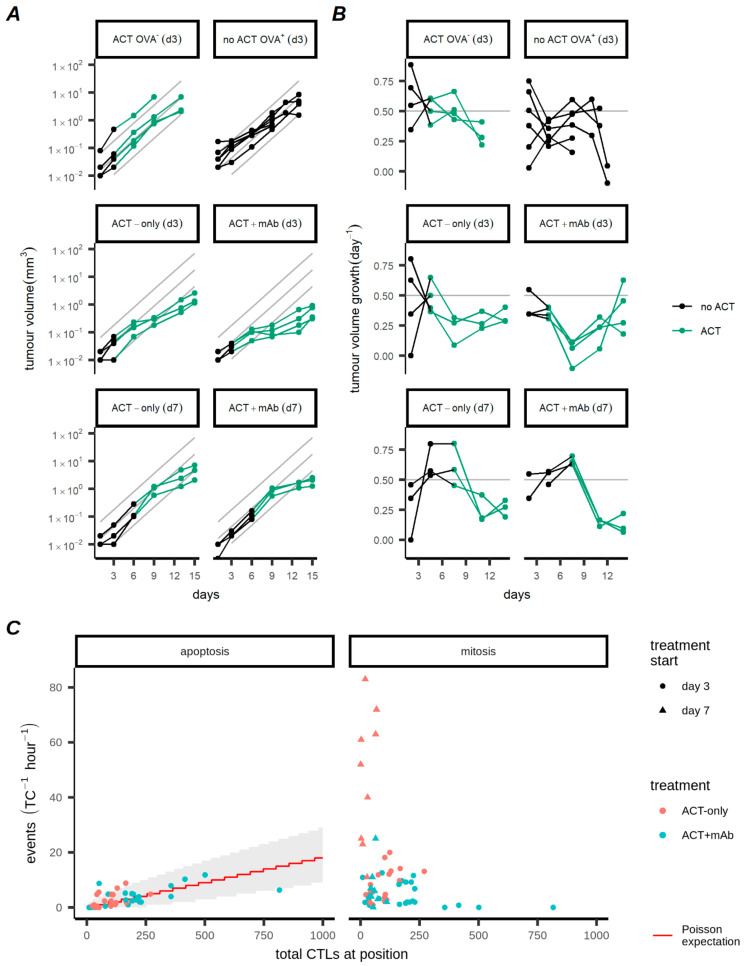
Impact of CTLs on the population dynamics of tumour cells. (**A**) Tumour volume measurements over time. Row 1: control tumours, either treated with ACT 3 days after inoculation but not expressing OVA (top left), or OVA expressing tumours not treated with ACT (top right). Rows 2 and 3: OVA expressing tumours treated with ACT 3 days (row 2) or 7 days (row 3) after inoculation, without anti-CD137 (left) or with anti-CD137 (right). (**B**) Volumetric growth rates of tumours corresponding to (**A**). Estimates of tumour growth rate are made over the interval between two successive volume measurements. Points representing estimates are displayed at the midpoint of the interval. Treatments for conditions are indicated in facet labels. Points in (**A**,**B**) are connected with straight lines visualizing the trajectories for individual mice. Points/lines corresponding to mice that had not (yet) received ACT are black, and green indicates that mice have received ACT. Grey lines in (**A**,**B**) represent the estimated growth rate of 0.5 day^−1^. (**C**) Relationship between the number of TC apoptosis events (left panel) or mitosis events (right panel) vs. number of CTLs per position. TC apoptosis and mitosis events are normalised (hour^−1^) to account for differences in imaging time between positions. The expected number of kills per hour (red line) and 5–95% confidence interval (shaded region) are shown for a Poisson process where individual CTLs kill at a constant rate (0.44 CTL^−1^ day^−1^). See also Appendix A for day 7 mitosis data.

**Figure 3 cancers-13-02567-f003:**
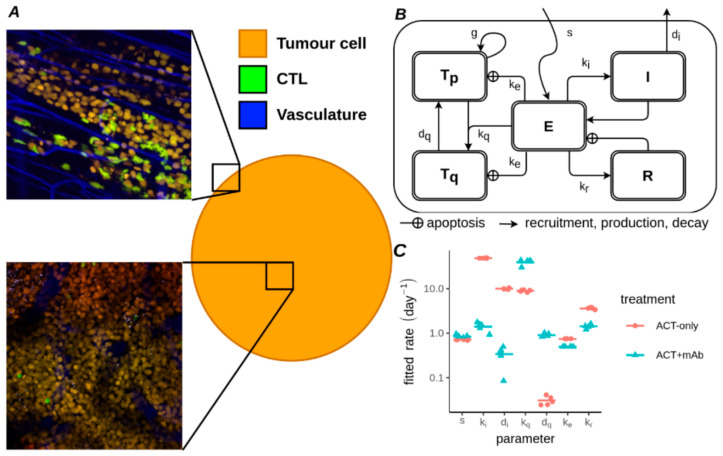
ODE model linking intravital and volumetric measurements from ACT treated tumours. (**A**) Examples of imaged positions with varying numbers of tumour cells, shown with their presumed location inside the tumour (circle). (**B**) Schematic of ODE model. (**C**) Best fitting parameters for the ODE model. Each point represents 1 of 5 fits using the stochastic evolutionary algorithm. Horizontal lines represent the mean fitted parameter for either ACT-only (red) or ACT+mAb (blue) conditions.

**Figure 4 cancers-13-02567-f004:**
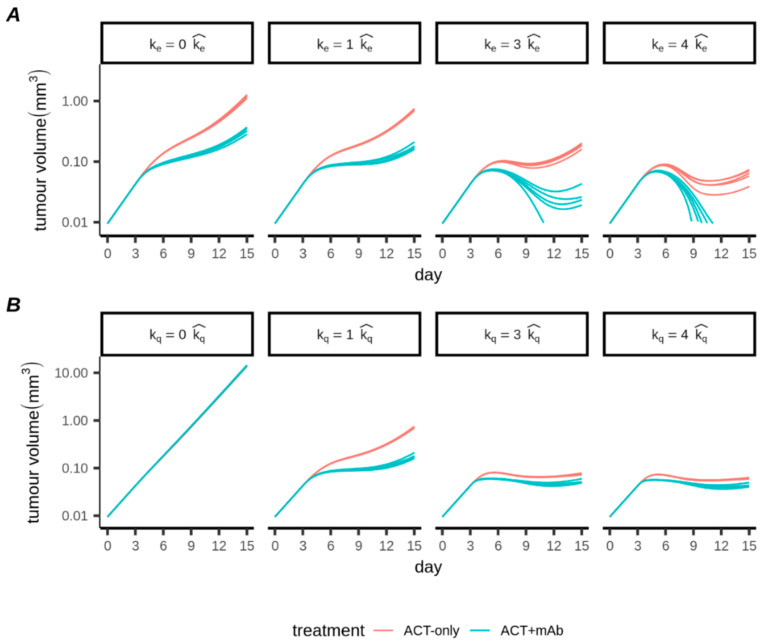
Relative impacts of antiproliferative and killing effects of adoptively transferred CTLs. (**A**,**B**) Impact of varying the killing rate (k_e_; **A**) and the rate parameter for induction of antiproliferative effect (k_q_; **B**). Using the best fitting parameter sets, both rates were multiplied by a factor of 0, 1, 3 or 4 (indicated by columns). Parameters with hats represent best fitting values.

**Figure 5 cancers-13-02567-f005:**
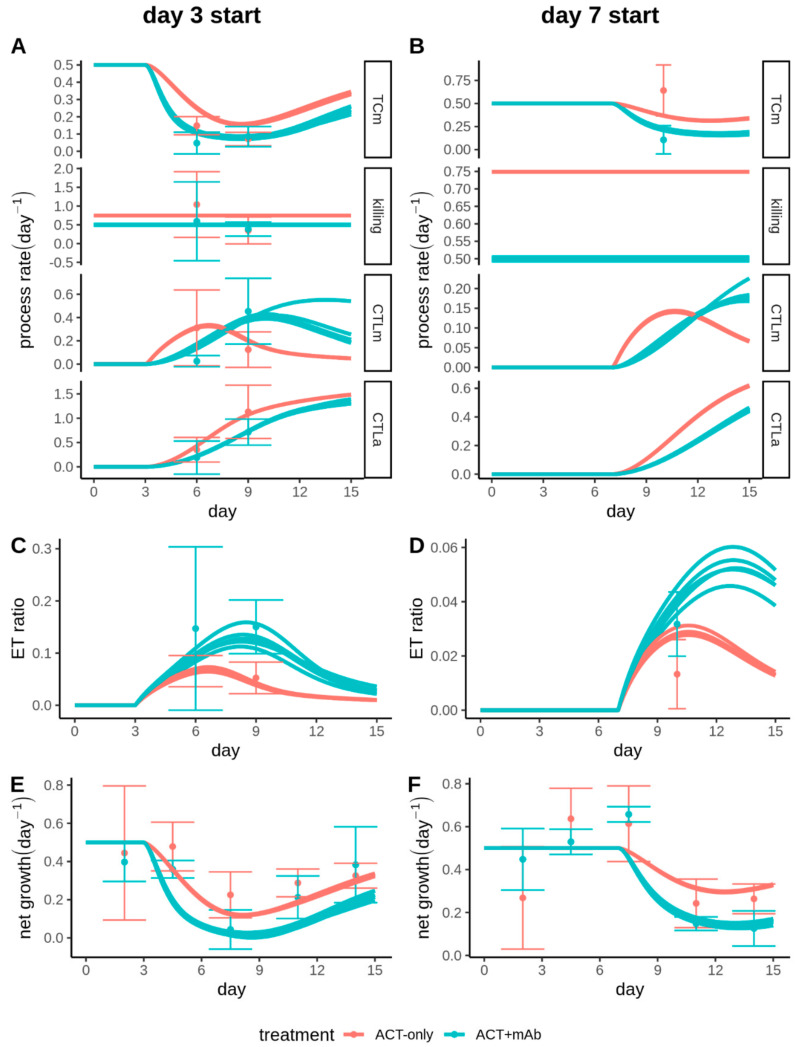
Enhanced antiproliferative effect and extended effector window of CD137 stimulated CTLs. (**A**–**F**) Results of parameter estimation. Model output represented by lines, data plotted as mean and s.d. Shown are observed and fitted process rates (**A**,**B**), effector:target ratio (**C**,**D**) and net tumour growth (**E**,**F**) for either tumours treated on day 3 (**A**,**C**,**E**) or day 7 (**B**,**D**,**F**). Process rates considered in (**A**,**B**) are: TCm (Tumour Cell mitosis); killing (of tumour cells by CTLs); CTLm (CTL mitosis); CTLa (CTL apoptosis).

**Table 1 cancers-13-02567-t001:** Calculation of intravital process rates from experimental data and from ODE model.

Process	Calculation from Experimental Data	Calculation from Model
CTL killing rate	(tumour cell apoptosis)/(time∙CTLs)	k_e_
Tumour cell mitosis rate	(tumour cell mitosis)/(time∙tumour cells)	gT_p_/T
CTL mitosis rate	(CTL mitosis)/(time∙CTLs)	I
CTL apoptosis rate	(CTL apoptosis)/(time∙CTLs)	R

**Table 2 cancers-13-02567-t002:** Best fitting model parameter values and their explanations.

Parameter Name	Explanation	ACT+mAb	ACT-only
g (day^−1^)	TC mitosis rate	0.5	0.5
s (TC^−2/3^ day^−1^)	Rate constant for CTL infiltration into tumour	0.87	0.7
k_e_ (CTL^−1^ day^−1^)	Rate at which CTLs kill tumour cells	0.5	0.75
k_i_ (day^−1^)	Rate of CTL-induced increase in CTL mitosis	1.3	49.6
d_i_ (day^−1^)	Rate of decrease in CTL-induced CTL mitosis	0.32	10
k_r_ (day^−1^)	Rate of CTL-induced CTL apoptosis (resistance).	1.4	3.8
k_q_ (day^−1^)	Rate CTLs induce antiproliferative effect	41.2	9.2
d_q_ (day^−1^)	Rate CTL induced antiproliferative effect disappears	0.97	0.03

## Data Availability

Source code for this work is available with the DOI:10.5281/zenodo.4443230.

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
