# Peer review of "Mathematical Modelling Based on In Vivo Imaging Suggests CD137-Stimulated Cytotoxic T Lymphocytes Exert Superior Tumour Control Due to an Enhanced Antimitotic Effect on Tumour Cells"

_cancers, 2021, doi:10.3390/cancers13112567_

Round 1

Reviewer 1 Report

Unfortunately, I don't find the authors’ answers to my previous comments satisfactory. I still believe that the model is flawed in its core and the argument that it works in the studied regime is not convincing (derivation from the double saturation model did not solve the issue). It is especially important since the authors present results in which there is a substantial change in tumor volume (Fig 4A). Moreover, it is known from biology that direct recognition of tumor cell is required to stimulate CTLs and missing interaction term is problematic. Finally, extravasation at the tumor site happens regardless of ability of CTL to recognize tumor cells (it is random + inflammation related) and thus there should be some amount of CTLs at the tumor site in case of OVA was absent (also after extravasation CTLs are not leaving of antigen is not recognized).

Author Response

We understand the concerns of this reviewer with respect to the utilized killing term. Nevertheless, we feel that it is better to utilize the simplified killing term in this case, because the limited data do not allow to determine all parameters of the alternatively possible double saturation function to describe CTL-mediated killing. To further acknowledge the limits of our model, in the revised version we highlighted the regime in which our killing rate approximation is valid (lines 198-202). Moreover, we emphasize the fact that our model may overstate the importance of killing when the killing rate is increased (as we do in Figure 4), due to the approximation made for the killing term (lines 491-494).

Reviewer 2 Report

I thank the authors for addressing my previous comments. I think the manuscript is now much stronger and clearer.

I don't have any further suggestions for changes to make at this stage. 

Author Response

We thank the reviewer for this positive evaluation.

This manuscript is a resubmission of an earlier submission. The following is a list of the peer review reports and author responses from that submission.

Round 1

Reviewer 1 Report

The authors propose an ODE model which, after fitting to experimental data, is aimed at answering two main questions: a) what is the most prominent mode of action of cytotoxic T cells (CTLs) against the tumor and b) what is the main mechanism governing an enhanced tumor control when CD137 targeted antibody is added together with CTLs. The authors consider two main modes of action: 1) direct T-cell mediated killing of tumor cells and 2) antiproliferative effect of factors secreted by T-cells. The model was fitted to the experimental data from murine model in which OVA specific T-cells where transferred to mice implanted with tumors transfected with OVA antigen. The main finding is (in addition to interesting data analysis) that estimated model parameters and numerical solutions to the model imply that T cells mainly exert antiproliferative effect on the tumor cells (what is additionally augmented by CD137 antibody) and direct killing is very limited.

I found the paper interesting, well written and tackling an interesting problem. It is a very nice example of how mathematical modeling can augment our understanding of biological data. There is, however, one major problem that I have with the model, which needs to be addressed before the paper is acceptable for publication.

Namely, the argument that the authors use about the lack of evidence for CTLs mediated tumor killing being dependent on tumor cells number is not only not convincing, but also makes the ODE model faulty. It is quite evident that in absence of tumor cells there can’t be any killing – the authors simply observed in the data cases in which the environment was always rich in tumor cells. By assuming, in contrary to the vast amount of literature on ODE modeling of tumor-immune system interaction, that the term in the model describing tumor cells killing is only proportional to the number of effector cells, the model loose one crucial property: non-negativity of solutions. You can quickly realize that negative solutions can arise, if you look at the extreme case and assume that the tumor is fully quiescent, i.e. Tp = const = 0 and d_q = 0, then the right hand side of equation for dT_q/dt is simply -keE. Similar problem arises if you assume that there is no quiescence at all (kq=0 and Tq=0). That is exactly the reason why mass action term is necessary.

Moreover, I would like to see more argumentation on the specific form of equations for R and I. Shouldn’t the equation for R have some negative feedback? It is hard to imagine and biologically unplausible that its value is potentially unbounded. Also, as far as I understand the concept, the variable I should be dependent on the strength of the recognition of CTLs of tumor cells, i.e. there should be some CTLs-tumor cells interaction term in the equation.

Finally, I don’t find the argument that CD137 most likely improves tumor control via enhancement of the antiproliferative effect of CTLs convincing. This is because in Fig. S3 we can see that for largest value of g there is virtually no difference between parameter k_q between experiments with or without CD137 antibody (the difference appears for lower growth rates). However, there is a consistent difference in other parameters values – maybe they should be looked at when explaining the CD137 mode of action?

Minor issues:

  • I would like to see the quality of fit of the model to the volume data and not only the growth rates.
  • One of the experiments considered implanting tumor cells without OVA antigen. This should be the case when there is no cell killing, but only the antiproliferative effect (under the model assumptions it is not dependent on direct interaction and recognition). How does the model compare to the data in this case? Should we still expect substantial tumor control?
  • Is the assumption that the tumor is spherical really valid in this specific experimental setting? The window is quite slim. If not, one should consider different CTL inflow term in the model.
  • Please don’t use the partial derivative operator in the equations, because you use ODEs. Just use dT/dt.
  • What are the initial conditions for the model?
  • In Figure 1E please use d9 – d6 to avoid confusion.
  • Could the authors provide more details on utilized optimization method and its specific settings?

Author Response

The authors propose an ODE model which, after fitting to experimental data, is aimed at answering two main questions: a) what is the most prominent mode of action of cytotoxic T cells (CTLs) against the tumor and b) what is the main mechanism governing an enhanced tumor control when CD137 targeted antibody is added together with CTLs. The authors consider two main modes of action: 1) direct T-cell mediated killing of tumor cells and 2) antiproliferative effect of factors secreted by T-cells. The model was fitted to the experimental data from murine model in which OVA specific T-cells were transferred to mice implanted with tumors transfected with OVA antigen. The main finding is (in addition to interesting data analysis) that estimated model parameters and numerical solutions to the model imply that T cells mainly exert antiproliferative effect on the tumor cells (what is additionally augmented by CD137 antibody) and direct killing is very limited. I found the paper interesting, well written and tackling an interesting problem. It is a very nice example of how mathematical modeling can augment our understanding of biological data.

We would like to thank the reviewer for acknowledging that our approach is a very nice example of the strength of mathematical modeling for understanding biological data.

There is, however, one major problem that I have with the model, which needs to be addressed before the paper is acceptable for publication. Namely, the argument that the authors use about the lack of evidence for CTLs mediated tumor killing being dependent on tumor cells number is not only not convincing, but also makes the ODE model faulty. It is quite evident that in absence of tumor cells there can’t be any killing – the authors simply observed in the data cases in which the environment was always rich in tumor cells. By assuming, in contrary to the vast amount of literature on ODE modeling of tumor-immune system interaction, that the term in the model describing tumor cells killing is only proportional to the number of effector cells, the model loose one crucial property: non-negativity of solutions. You can quickly realize that negative solutions can arise, if you look at the extreme case and assume that the tumor is fully quiescent, i.e. Tp = const = 0 and d_q = 0, then the right hand side of equation for dT_q/dt is simply -keE. Similar problem arises if you assume that there is no quiescence at all (kq=0 and Tq=0). That is exactly the reason why mass action term is necessary.

The reviewer is correct that our simplification is not applicable in all regimes. Indeed, the simplification that killing is only proportional to number of CTLs would of course break down for extremely low tumour cell densities, when the time required for CTLs to locate new targets would become the most important determinant of killing rate. However, we argue that our simplification is valid for those tumours we described here since such low tumour cell densities are never reached. In fact, in our model we take the tumour cell density to be constant which we believe to be justified here because the measured apoptosis rates of tumour cells are low whilst the measured mitosis rates of tumour cells are relatively high. Together these factors mean that regions of killed tumour cells do not accumulate quickly and can soon be refilled by dividing tumour cells. This means that, from the perspective of the CTLs there are always sufficient available tumour cell targets and therefore CTLs can kill at a constant rate. We have clarified this issue in the revised version by taking our earlier published dual saturation model as a starting point (Gadhamsetty et al., 2014), and then explaining our rationale for reducing the model to a killing term which is linear in CTL numbers. Moreover, we have modified figure 4A (where we simulate higher kill rates by CTLs) by terminating our simulations earlier in order to make clear that our model assumptions do not remain valid in such a regime where rapid destruction of the tumours is occurring. 

Moreover, I would like to see more argumentation on the specific form of equations for R and I. Shouldn’t the equation for R have some negative feedback? It is hard to imagine and biologically unplausible that its value is potentially unbounded.

We agree with the reviewer that the value of model variables representing biological entities should in general not be unbounded. In the current model, R does have negative feedback in the sense that as R increases, the number of effector cells E will decrease, which will in turn stop further increases to R. In a previous model version we had also looked at the possibility of adding negative feedback explicitly into the equation for R, i.e.:

dRdt=krE-dRR

but fitting resulted in very small values for the parameter dR. We reasoned that there was thus no meaningful decrease in resistance on the (relatively short) timescale over which the experimental observations were made. At a long timescale a decrease in resistance is very likely to occur, but based on the current data we cannot quantify this effect, and we thus feel we should keep the model as simple as possible. We have included additional text in the Methods to clarify this point. 

The variable I should be dependent on the strength of the recognition of CTLs of tumor cells, i.e. there should be some CTLs-tumor cells interaction term in the equation.

Indeed, the variable I could depend on CTL recognition of tumour cells, however this should only be necessary to include explicitly in the event that the recognition of tumour cells is variable over the period of the experiments. Using similar reasoning as for the killing term, we consider that the effect of the tumour cells is in a saturated regime. We think that this is reasonable for the current experimental data for which CTLs continuously experience a very high local tumour cell density and where only little killing occurs. In our revised submission we have included this reasoning, and acknowledge that this might be modeled differently in other settings (referring to a paper where this was indeed done differently). 

I don’t find the argument that CD137 most likely improves tumor control via enhancement of the antiproliferative effect of CTLs convincing. This is because in Fig. S3 we can see that for largest value of g there is virtually no difference between parameter k_q between experiments with or without CD137 antibody (the difference appears for lower growth rates). However, there is a consistent difference in other parameters values – maybe they should be looked at when explaining the CD137 mode of action?

We thank the reviewer for pointing this out. In fact, the range of values for the tumour growth rate in the absence of CTLs that we presented in our previously submitted version was simply too large, as some of the values would not at all result in a good match to the experimental data for the untreated tumours and thus was an irrelevant parameter regime to consider. In the revised version we have included an additional figure (Fig. S2) where we show comparisons of the untreated tumour growth data with different growth rates to illustrate this point. Moreover, we have removed the unrealistic values of the tumour growth rate (g) from our previous Fig. S3 (now Fig. S6) to focus only on the relevant range.

I would like to see the quality of fit of the model to the volume data and not only the growth rates.

We have included an additional figure in the supplement to show how our model fitted on the basis of the intravital imaging measurements matches the volumetric growth data (Fig. S4). Note that this match is relatively good, although the model slightly underpredicts growth at late time points for the case where CTLs are transferred on day 3. 

One of the experiments considered implanting tumor cells without OVA antigen. This should be the case when there is no cell killing, but only the antiproliferative effect (under the model assumptions it is not dependent on direct interaction and recognition). How does the model compare to the data in this case? Should we still expect substantial tumor control?

Our model was parameterised for the situation where OVA was present, thus we would not necessarily expect any of our parameters to remain valid for the tumours without OVA. For example, the parameter s represents the net infiltration of CTLs into the tumours. However, if the CTLs do not recognise the tumour cells (as would be expected for the transferred CTLs, as they only recognize a specific OVA epitope), then there would be no arrest of CTLs inside the tumour and they should leave just as quickly as they arrive. Moreover, as discussed above the parameter ki already includes the strength of interaction between CTLs and the tumour - it is likely that we would find a value close to 0 for ki in a tumour without OVA. Finally, both parameters ke and kq should be 0 in this case as without antigen we would not expect any effector function. In our revised submission we have clarified this by including a more thorough description of the interpretation of our parameters to highlight how we would expect them to change in situations where the strength of recognition of the tumour varied.  

Is the assumption that the tumor is spherical really valid in this specific experimental setting? The window is quite slim. If not, one should consider different CTL inflow term in the model.

The assumption is valid to the best of our knowledge. At the moment the mice were sacrificed the tumours were still quite small, thus preventing the window unduly influencing the growth of the tumours (note in Fig. 2b an upper bound on tumour volumes of approximately 10 mm3). We have included additional text in the methods section to clarify this point.

Please don’t use the partial derivative operator in the equations, because you use ODEs. Just use dT/dt.

We apologise for this inaccuracy and corrected it in our revised version.

What are the initial conditions for the model?

We thank the reviewer for noticing that the initial conditions were not mentioned in the previously submitted version. We have now detailed these in subsection 2.2.4 of our revised draft.

In Figure 1E please use d9 – d6 to avoid confusion.

We thank the reviewer for pointing this out and we have made this modification on our revised draft. 

Could the authors provide more details on utilized optimization method and its specific settings?

We have included additional detail in the final paragraph of the methods section which now includes the permitted bounds for each parameter in the optimisation procedure - this paragraph should now contain a complete description required to reproduce the optimisation. Moreover, we have also added some additional explanation of our optimisation procedure in the same paragraph.

Reviewer 2 Report

In this paper, a mathematical model is developed and applied to murine data, describing treatment via cytotoxic T cells with or without anti-CD1237, in order to control tumour dynamics. The authors find that CD137 may enhance the antiproliferative effect of tumours, specifically in melanoma. This could provide experimental possibilities in the future on tumour control. I would recommend the acceptance of the paper after considering a discussion of the following comments:

  1. In relation to the model presented, there are several bibliography references considering mathematical models which include T-cell interaction and tumour control. Which are the main differences and novelties, mathematically speaking, included on the model proposed? Please provide biological motivation or reasoning beyond the models already present in the literature. I would suggest elaborating more on the possibility of including human data, in order to complement the murine data provided.
  2. Considering the parameter selection, I would suggest a more detailed analysis, specifically to examine a wider parameter range besides those fixed values shown in Table 2. This would allow to show whether low/high changes on the parameters have an influence on the dynamics of the model, and hence, whether the conclusions obtained differ.
  3. In regard to scientific soundness, I was expecting a fitting of the CTL number by means of simulation of the solution of Equation (4). However, only two temporal data points are used in several simulations, which is problematic in terms of fitting. In fact, any function could be fitted to two points (see Fig. 5A-D or Fig. S3A-B and S3C). Could this limitation be amended? If the temporal data were available, do you think the model parameters in current setting would be appropriate to be fitted?
  4. Considering presentation, I would suggest that a higher quality of the Figures may provide a better understanding of the processes described. For example, legends can be difficult to understand: icons on Fig. 1A are different on the legend than in the graphs, point size in Fig. 1C do not show big differences, etc. In several Figures there are objects which are not described neither in the captions nor in the main text (light grey lines in Fig 2A, parameters k (with hat) in Fig. 4, sign + with circle in Fig 3B, meaning of the intensity of colour in Fig 3A, etc.). Even if these changes are made, I suggest a deeper explanation and/or higher quality of the figures to enhance rigorousness. 
  5. The results would need to be clearer, as the main conclusion is sparsed along all subsections in Section 3. I would suggest to include shorter, preciser conclusions in the main text. For example it would be benefitial to also include the conclusion obtained in each subsection title, so that the reader could better understand where the the paper is headed to. I consider that this would attract a wide readership. Finally, subsection 3.3 seems more methodological than a result.

Author Response

In relation to the model presented, there are several bibliography references considering mathematical models which include T-cell interaction and tumour control. Which are the main differences and novelties, mathematically speaking, included on the model proposed? Please provide biological motivation or reasoning beyond the models already present in the literature. 

We found that previous models were not appropriate to apply to our data, because they were either over parameterised or would not fit the additional data we had concerning intravital measurements of cellular apoptosis and mitosis rates (e.g. many existing models assume a constant death rate of CTLs whereas it was clear in our data that this rate was increasing over time).  We have clarified this point in our revised manuscript and we have added some additional discussion where we more explicitly compare our model to pre existing models in the literature. 

I would suggest elaborating more on the possibility of including human data, in order to complement the murine data provided.

As suggested by the reviewer, we have added additional discussion about how we might extend our models to human data. 

Considering the parameter selection, I would suggest a more detailed analysis, specifically to examine a wider parameter range besides those fixed values shown in Table 2. This would allow to show whether low/high changes on the parameters have an influence on the dynamics of the model, and hence, whether the conclusions obtained differ.

The main purpose of the model was to apply it to the experimental data in order to understand and quantify the processes inside these in vivo tumours, rather than to explore the model very much from a mathematical viewpoint.  Changes to the model parameters would result in a model which does not fit well to the experimental data. To illustrate this, we have included a sensitivity analysis to show how changing the model parameters relative to the best fit decreases the RMSE. 

In regard to scientific soundness, I was expecting a fitting of the CTL number by means of simulation of the solution of Equation (4). However, only two temporal data points are used in several simulations, which is problematic in terms of fitting. In fact, any function could be fitted to two points (see Fig. 5A-D or Fig. S3A-B and S3C). Could this limitation be amended? If the temporal data were available, do you think the model parameters in current setting would be appropriate to be fitted?

The reviewer is correct that two data points are insufficient to constrain a function. However, since the model is simultaneously fit to all the data we have, then there are in fact additional constraints. In particular, the volumetric growth data has 5 data points. Since we see that the tumour growth rate starts high (days 0-6) and ends high (days 12-15), this indirectly constrains the number of CTLs inside the tumour since if there were too many CTLs then a greater reduction in the tumour growth rate should occur. We have included additional explanation in the methods to clarify this, and we also mention the limited number of temporal data points in our discussion and the benefit of obtaining additional time points in the future.   

Considering presentation, I would suggest that a higher quality of the Figures may provide a better understanding of the processes described. For example, legends can be difficult to understand: icons on Fig. 1A are different on the legend than in the graphs, point size in Fig. 1C do not show big differences, etc. In several Figures there are objects which are not described neither in the captions nor in the main text (light grey lines in Fig 2A, parameters k (with hat) in Fig. 4, sign + with circle in Fig 3B, meaning of the intensity of colour in Fig 3A, etc.). Even if these changes are made, I suggest a deeper explanation and/or higher quality of the figures to enhance rigorousness

We have corrected the legend in Fig. 1A, and added some explanation of the point size in the text corresponding to Fig. 1C. The light grey lines in Fig. 2A were described in the text already, but in the revised manuscript we have also included this description in the figure legends to improve clarity. We also rectified the issues with Fig. 3A, 3B and Fig. 4 mentioned by the reviewer. Finally we have made various modifications to the figures to improve clarity and presentation throughout the manuscript. 

The results would need to be clearer, as the main conclusion is sparsed along all subsections in Section 3. I would suggest to include shorter, preciser conclusions in the main text. For example it would be benefitial to also include the conclusion obtained in each subsection title, so that the reader could better understand where the the paper is headed to. I consider that this would attract a wide readership. Finally, subsection 3.3 seems more methodological than a result.

We have modified the titles of subsections 3.1 & 3.2 to reflect the conclusions reached in these subsections, and improved the clarity of the conclusions per subsection of the Results. Moreover we have slightly restructured subsection 3.3 to focus on the result here (which is that the model we developed is able to describe our data).